# The Role of CCL24 in Primary Sclerosing Cholangitis: Bridging Patient Serum Proteomics to Preclinical Data

**DOI:** 10.3390/cells13030209

**Published:** 2024-01-23

**Authors:** Raanan Greenman, Tom Snir, Avi Katav, Revital Aricha, Inbal Mishalian, Ophir Hay, Matthew Frankel, John Lawler, Francesca Saffioti, Massimo Pinzani, Douglas Thorburn, Amnon Peled, Adi Mor, Ilan Vaknin

**Affiliations:** 1Chemomab Therapeutics Ltd., Tel Aviv 6158002, Israel; 2Goldyne Savad Institute of Gene Therapy, Hebrew University Hospital Jerusalem, Jerusalem 91120, Israel; 3UCL Institute for Liver and Digestive Health, University College of London, London NW3 2PF, UK; 4Sheila Sherlock Liver Centre, Royal Free London NHS Foundation Trust, London NW3 2QG, UK; 5Department of Gastroenterology and Hepatology, Oxford University Hospitals NHS Foundation Trust, Oxford OX3 9DU, UK

**Keywords:** cholangitis, inflammation, fibrosis, CCL24, chemokines, proteomics, hepatic stellate cells, neutrophils, monocytes

## Abstract

Primary sclerosing cholangitis (PSC) is an inflammatory and fibrotic biliary disease lacking approved treatment. We studied CCL24, a chemokine shown to be overexpressed in damaged bile ducts, and its involvement in key disease-related mechanisms. Serum proteomics of PSC patients and healthy controls (HC) were analyzed using the Olink^®^ proximity extension assay and compared based on disease presence, fibrosis severity, and CCL24 levels. Disease-related canonical pathways, upstream regulators, and toxicity functions were elevated in PSC patients compared to HC and further elevated in patients with high CCL24 levels. In vitro, a protein signature in CCL24-treated hepatic stellate cells (HSCs) differentiated patients by disease severity. In mice, CCL24 intraperitoneal injection selectively recruited neutrophils and monocytes. Treatment with CM-101, a CCL24-neutralizing antibody, in an α-naphthylisothiocyanate (ANIT)-induced cholestasis mouse model effectively inhibited accumulation of peribiliary neutrophils and macrophages while reducing biliary hyperplasia and fibrosis. Furthermore, in PSC patients, CCL24 levels were correlated with upregulation of monocyte and neutrophil chemotaxis pathways. Collectively, these findings highlight the distinct role of CCL24 in PSC, influencing disease-related mechanisms, affecting immune cells trafficking and HSC activation. Its blockade with CM-101 reduces inflammation and fibrosis and positions CCL24 as a promising therapeutic target in PSC.

## 1. Introduction

Primary sclerosing cholangitis (PSC) is a chronic liver disease without approved treatment, leading to end-stage liver disease and cholangiocarcinoma. It is characterized by ductular reaction, progressive fibrosis, and peribiliary inflammation involving T cells, neutrophils, and macrophages [1,2]. The etiology and pathogenesis of PSC remain unclear, presenting ongoing scientific and clinical challenges.

Chemokines are implicated in many chronic liver diseases, including PSC [3]. The contributions of various chemokines to liver disease and fibrosis are diverse and include canonical roles of modulating hepatic inflammation and directly recruiting resident liver cells, such as hepatic stellate cells (HSCs). In addition, non-canonical roles, such as cell proliferation and HSC activation, contribute to the aggravation of liver damage [4]. Better understanding of the functional contributions of chemokines and cell subsets to PSC progression can lead to potential targets for therapeutic intervention.

CCL24, a member of the C-C chemokine ligand family, binds to its sole receptor, CCR3, and plays a role in type 2 immune response and in fibrotic processes [5]. It is secreted by macrophages and epithelial cells and is involved in chronic inflammation and fibrosis in various organs, including the liver, skin, heart, and lung [6,7,8,9]. In liver diseases such as non-alcoholic steatohepatitis (NASH) and hepatocellular carcinoma (HCC), CCL24 and CCR3 are upregulated, and CCL24 blockade with the neutralizing antibody CM-101 reduced liver damage and fibrosis in preclinical models [6,10]. We previously demonstrated elevated expression of CCL24 and CCR3 in injured peribiliary areas [11]. CM-101 treatment in PSC preclinical models improved inflammation, fibrosis, and cholestasis-related markers along with reduced HSC activation and reduced infiltration of macrophages. In this study, we employed several experimental high-throughput methods to elucidate the underlying disease mechanisms in PSC driven by CCL24.

## 2. Results

### 2.1. Pathways Related to PSC and Its Severity Are Elevated in Patients with High CCL24 Levels

To investigate the involvement of CCL24 in PSC, we analyzed the serum proteome of two cohorts of patients with PSC and one cohort of healthy controls (HC) (characteristics of the study cohorts are outlined in Table 1). We profiled the expression of 2081 extracellular proteins using the Olink^®^ Explore high-throughput proteomic platform [12] (Figure 1A). By grouping together the two similar PSC cohorts (Appendix A and Table 1), we were able to obtain a more comprehensive representation of patients with PSC. Overall, patients with PSC had a distinctly altered proteome profile from HC, as demonstrated by principal component analysis of the study population (Figure 1B). Furthermore, stratifying patients with PSC by fibrosis severity (examined by Enhanced Liver Fibrosis (ELF) score with a 9.8 cutoff [13,14]) showed a distinct proteome profile in severe patients (Figure 1B).

To assess the involvement of CCL24 in PSC, we examined the association of CCL24 circulating levels with other proteins in patients with higher risk of disease progression (ALP above 1.5 of upper limit of normal (ULN) values). CCL24 was significantly correlated with serum proteins frequently associated with inflammation, chemotaxis, and fibrosis (Figure 1C and Appendix A) in patients with PSC but not in HC.

To unveil the mechanisms impacted in PSC and their relation to CCL24, we analyzed the data by three comparisons: by disease (patients with PSC vs. HC), by fibrosis severity (patients with PSC with high vs. low ELF score), and by CCL24 levels (patients with PSC with high vs. low serum CCL24 expression, demographics presented in Table 2) (Figure 1A and Appendix A). Differentially expressed proteins (DEPs) from each of the three comparisons were interpreted using ingenuity pathway analysis (IPA), to identify dysregulated canonical biological pathways, upstream regulators, and toxicity functions. The enriched canonical pathways, upstream regulators, and toxicity functions overlapped between the three comparisons (disease, fibrosis, and CCL24), suggesting that there are biological mechanisms related to PSC and its progression, which also relate to CCL24 expression levels (Figure 1D–I and Appendix A). Overlapping canonical pathways included HSC activation pathway, immune cell trafficking pathways (granulocyte and agranulocyte adhesion and diapedesis), and inflammation pathways (Th1 and Th2 activation, Th1 activation, and pathogen-induced cytokine storm). To evaluate the biological context of protein sets, expression of proteins that appear in a pathway set was averaged per individual. Patients with PSC show higher expression compared to HC. Within patients with PSC, those that had high levels of CCL24 also showed significantly higher average expression of these pathways (Figure 1E and Appendix A). When examining IPA’s upstream regulators that overlapped between patients with PSC, patients with high ELF, and patients with high CCL24 expression, we observed a predominant inflammatory-fibrotic signature (e.g., TNF, IFNγ, IL1β, TGFβ, and IL4), which was mostly elevated in patients with high CCL24 levels (Figure 1F,G, Appendix A). A similar overlap was observed for liver-related toxicity functions (Figure 1H,I and Appendix A). These findings highlight the involvement of CCL24 in pathways that play a pivotal role in the development and progression of PSC. We validated the comparison between HC and PSC individuals using an age-matched subset of the data (clinical characteristics in Appendix A). Pathway analysis showed upregulation of the same canonical pathways, upstream regulators, and toxicity functions (Appendix A).

The eotaxins family consists of three chemokines, CCL24 (eotaxin-2), CCL11 (eotaxin-1), and CCL26 (eotaxin-3), that signal via the same receptor, CCR3. This may be interpreted as redundancy or a tissue- and expression-dependent high degree of specificity [15]. To decipher whether all three eotaxins are equally associated with PSC-related mechanisms, we first examined the correlation between them (Figure 2A) and their relation to the ELF score. CCL24 and CCL11 were correlated specifically in patients with PSC, but not in HC, whereas CCL26 did not correlate with either CCL24 or CCL11. When comparing high ELF-score to low ELF-score patients, no significant difference was observed in eotaxin expression levels (Appendix A). Next, we examined whether eotaxin expression levels were associated with an elevated degree of the aforementioned pathways, upstream regulators, and toxicity functions (Figure 2B and Appendix A). A unique association of CCL24 expression levels, and not CCL11 or CCL26, with PSC-related mechanisms implied specific, non-redundant involvement of CCL24 in PSC.

### 2.2. An In Vitro-Based CCL24-Dependent Protein Signature Differentiates PSC and Its Severity

As shown above, proteomic analysis revealed key pathways, such as HSC activation and immune cell trafficking, that are implicated in PSC and in patients with elevated CCL24 levels. Neutralizing CCL24 in experimental models of liver damage affected HSCs and immune cells [6,7,11], prompting us to further explore the role of CCL24 within these pathways. HSC activation by CCL24 was previously demonstrated by HSC proliferation, migration, and ECM production [6,11]. To gain more insight on HSC activation by CCL24, we screened the secretome of the HSC line LX2 that was activated with CCL24 or blocked with CM-101 (Figure 3A,B). CCL24-activated HSCs increased secretion of pro-fibrotic pro-inflammatory proteins associated with HSC activation, including IL1β, TIMP1, and leptin (Appendix A). Furthermore, we generated a CCL24-dependent protein signature based on proteins that exhibited a twofold change in expression upon CCL24 treatment, with their expression levels returning to control levels upon CCL24 blockade with CM-101 (Figure 3C). We then examined whether this CCL24-dependent secretome profile can reflect individuals’ disease state. Differences in the serum proteomic profile of the CCL24-dependent signature are shown as a clustered heatmap (Figure 3D). Generally, HC were grouped together and patients with PSC were grouped by fibrosis severity. Similarly, mean expression of proteins within the signature is higher in patients and in those with severe fibrosis (Figure 3E,F). To evaluate individual proteins that highly contribute to differentiating disease and fibrosis severity, receiver operating characteristic (ROC) analysis was performed for the CCL24-dependent signature. Several proteins predicted disease and fibrosis states with area-under-curve values of >0.9 and >0.8, respectively, while their combination resulted in an even greater predictive capacity (Figure 3G,H). Altogether, this differentiation based on the CCL24-dependent protein signature reinforces the pro-fibrotic role of CCL24 in PSC.

### 2.3. Enhanced Local Neutrophil and Monocyte Recruitment Following CCL24 Administration

Based on the association of CCL24 with immune cell adhesion and diapedesis pathways, we aimed to identify the specific immune subtypes recruited by CCL24. To achieve this, we utilized a murine model of immune trafficking, administering intraperitoneal injections of PBS, CCL24, or CCL11 (Figure 4A). Immune cell composition using single-cell RNA sequencing (scRNA-seq) identified 10 immune cell types (Figure 4B,C and Appendix A). CCL24 injection induced distinct changes in the immune cell compartment, characterized by recruitment of neutrophils, monocytes, and natural killer (NK) cells (Figure 4D and Appendix A). These changes were not observed in CCL11-treated animals. Additionally, CCL24 and CCL11 injections altered the activation/polarization state of peritoneal macrophages, leading to an increased accumulation of M2-like macrophages (Figure 4D and Appendix A). Moreover, CCL24 was found to be robustly expressed by multiple immune cell types in the peritoneum, particularly monocytes, macrophages, B cells, and dendritic cells (Figure 4E). Notably, CCL24 administration further induced expression of CCL24, as shown by the increased number of CCL24-positive cells (Figure 4F).

To corroborate these changes, we conducted further experiments employing the CM-101-neutralizing antibody or an isotype control. Fluorescence-activated cell sorting (FACS) analysis revealed an increase in Ly6c^hi^ monocytes and Ly6g^+^ neutrophils (Figure 4G–J). These changes were specific to CCL24 treatment, as opposed to CCL11 injection, and were effectively inhibited by subcutaneous injection of CM-101.

### 2.4. Treatment with CCL24-Neutralizing Antibody Attenuates Liver Fibrosis and Reduces Neutrophil and Macrophage Accumulation in Experimental Cholestasis Murine Model

Next, we used the ANIT-induced chronic cholestasis mouse model to examine the effect of CCL24 blockade with CM-101. Mice were fed with an ANIT diet for 4 weeks and treated twice a week with either 5 mg/kg CM-101 or a vehicle control (PBS) during weeks 2 through 4 (Figure 5A). Whereas ANIT-fed mice had a prominent increase in serum bile acid levels, treatment with 5 mg/kg CM-101 reduced serum bile acid levels by 43% in ANIT-fed mice (Figure 5B). Liver histological H&E staining was used to assess fibrosis, biliary hyperplasia, and necrosis (Figure 5C). CCL24 blockade ameliorated liver condition, evidenced by reduced liver fibrosis (Figure 5D), biliary hyperplasia (Figure 5E), and liver necrosis (Figure 5F). Based on the specific enhancement of neutrophils and monocytes following i.p. CCL24 injection, we performed immunological staining to examine peribiliary accumulation of neutrophils and macrophages compared with T cells. Treatment with CM-101 reduced accumulation of neutrophils and macrophages by 44% and 20%, respectively, in ANIT-fed mice compared to ANIT-fed mice treated with vehicle (Figure 5G,H). No significant T cell peribiliary accumulation difference was observed (Figure 5I).

To summarize the observed hepatic changes in relation to disease severity, we evaluated the correlation between fibrosis, biliary hyperplasia, liver macrophage accumulation, and neutrophil accumulation. High correlations are observed between these markers (Appendix A). Of these, neutrophil accumulation was strongly correlated with fibrosis and biliary hyperplasia grades (Spearman’s correlations of 0.70 and 0.78, respectively).

To corroborate these CCL24-dependent inflammatory modifications, we examined the Mdr2^−/−^ mice model for sclerosing cholangitis. We recently reported that CCL24 blockade reduced biliary damage in these mice, demonstrated by reduced fibrosis, reduced cholangiocyte and HSC proliferation, and reduced macrophage accumulation [11]. Further examining neutrophil accumulation by CXCR2 immunohistochemistry showed strong reduction compared with untreated Mdr2^−/−^ mice (Appendix A). Moreover, the correlation between disease markers (serum ALP, liver fibrosis, and cholangiocyte proliferation) and macrophage and neutrophil accumulation demonstrated a strong relation between cholangiocyte, macrophage, and neutrophil accumulation (Appendix A). To conclude, CCL24 blockade in PSC animal models demonstrated anti-fibrotic, anti-inflammatory effects, interfering with migration of monocytes and neutrophils to the damaged biliary area.

### 2.5. Neutrophil and Monocyte Migration Is Associated with PSC and CCL24 Levels

After establishing the role of CCL24 in peribiliary neutrophil and monocyte recruitment, we investigated their clinical manifestation in PSC. Recent studies identified neutrophil and monocyte peribiliary accumulation as markers for PSC disease [16,17]. To cross-validate the link between CCL24 and neutrophil and monocyte chemotaxis in patients, we analyzed serum proteomics by evaluating protein signatures of monocyte- and neutrophil-related pathways. Heatmaps illustrate an increased expression of these proteins in patients with PSC (Figure 6A,B) as compared to HC. Similarly, elevated levels of pathways associated with neutrophil recruitment were observed in patients with PSC, particularly in those with high CCL24 levels. Interestingly, these pathways were not elevated in patients with high CCL11 or CCL26 levels (Figure 6C,D). Moreover, pathways related to monocyte and monocyte recruitment were upregulated in PSC patients and were associated with CCL24 levels (Figure 6E,F). Collectively, these results underscore the activation of monocyte and neutrophil recruitment in patients with PSC, emphasizing its association with CCL24.

## 3. Discussion

This study investigated the involvement of CCL24 in PSC and its association with disease-related pathways. We further demonstrated that CCL24 directly contributes to the development of fibrosis and inflammation by activating HSCs and recruiting monocytes and neutrophils. Additionally, we showed the therapeutic potential of CCL24 blockade in a preclinical cholestasis model. These findings establish CCL24 as a central factor orchestrating fibrosis and inflammation in PSC progression, corroborating previous research that emphasizes the therapeutic efficacy of neutralizing CCL24 in preclinical PSC models [11].

Serum proteomics has emerged as a valuable tool in exploring the complex molecular landscape of chronic cholangiopathies, such as PSC. It allows us to gain insights into disease-related pathways, potential biomarkers, and our understanding of the underlying mechanisms driving cholangiopathies [18,19,20,21]. We showed the association of CCL24 with pivotal elements of PSC, such as immune cell recruitment, fibrosis and wound healing, Th1 and Th2 activation, and signaling through cardinal regulators, such as IFNγ, IL1β, and IL4. Indeed, the analysis predicted known regulators influencing the CCL24/CCR3 axis, such as IL4, IL10, STAT3, and IFNγ [22,23,24].

HSC activation is a key event in the development of hepatic fibrosis, tightly programmed and initiated by several factors, such as oxidative stress, extracellular matrix remodeling, and profibrotic cytokines and chemokines [4,25]. CCL24 induces HSC activation, including its proliferation and migration [6,11]. By profiling the secretome of CCL24-activated HSCs, we uncovered crucial secreted regulators that can exacerbate PSC. Moreover, this CCL24-dependent protein signature stratified patients based on disease presence or severity. The identification of a small subset of proteins, capable of distinguishing disease and severity levels, offers potential biomarkers to monitor therapeutic interventions in patients with PSC. Future investigations can explore the utility of this signature to assess HSC activation using noninvasive serum measurements or potentially serve as a biomarker to monitor treatment responses in anti-CCL24 therapies.

While CCL24 is a well-known eosinophil chemoattractant, its role in recruiting other immune cell types is understudied. Increased macrophage and neutrophil accumulation was reported in CCL24-rich skin tissues [26], mirroring our findings in liver and peritoneum tissues. Accumulation of peribiliary monocyte-derived macrophages was shown to be a feature of PSC, including M2-like macrophages, which promote fibrosis and secrete CCL24 [11,27,28]. Disruption of monocyte recruitment attenuated fibrosis and cholestasis [27,29]. Similarly, preclinical models demonstrated reduced macrophage accumulation in the liver following treatment with the CCL24-blocking antibody, CM-101 [6,11]. Recently, Sun et al. revealed that CCR3^+^ monocyte-derived macrophages mediate chronic liver injury and inflammation, which was restored upon CCR3 blockade [30]. While CCR3 binds multiple eotaxins, our proteomic analysis highlights the unique association of CCL24 with PSC mechanisms, with no similar association of CCL11 and CCL26, suggesting it as a key regulator of CCR3^+^ macrophage recruitment and emphasizing its potential as a therapeutic target.

Peribiliary infiltration of neutrophils is a common histological feature in chronic liver diseases, associated with increased liver injury, fibrosis, cholestasis, and portal hypertension [16,31]. Their recruitment can be driven by hepatic T cells and cholangiocytes [32]. In this study, we showed the direct impact of CCL24, primarily secreted by cholangiocytes and liver macrophages. We demonstrated reduced peribiliary neutrophil accumulation following CCL24 blockade. Blocking of neutrophil recruitment reduces ductular reaction, fibrosis, and angiogenesis [33], however, the precise mechanisms by which neutrophils contribute to PSC are yet to be fully elucidated.

The in vivo investigations conducted in this study highlight the mobilization of myeloid cells, specifically monocytes and neutrophils, without demonstrating a discernible impact on lymphoid cells. Nonetheless, the proteomic analysis revealed heightened activation of both Th1 and Th2 pathways in patients with elevated CCL24 levels, implying that the murine model might not capture the full scope of CCL24’s mode of action in the pathogenesis of PSC. 

In conclusion, our study provides further support for the involvement of CCL24 in PSC severity and disease-related pathways. The identification of a CCL24-related signature in HSCs, the specific recruitment of neutrophils and monocytes by CCL24, and the therapeutic benefits of CCL24 blockade contribute to our understanding of the pathogenesis of PSC. Targeting CCL24 could hold promise as a potential therapeutic strategy to ameliorate PSC progression and improve patient outcomes.

## 4. Conclusions

CCL24 is associated with PSC-related pathways.A CCL24-dependant signature observed in CCL24-treated hepatic stellate cells can differentiate patients with PSC by disease severity.CCL24 specifically recruits neutrophils and monocytes.CCL24 blockade reduces fibrosis, biliary hyperplasia, and neutrophil and monocyte recruitment.These data support further clinical investigation of an anti-CCL24 therapy for PSC.

## 5. Patients and Methods

### 5.1. Subjects

This study contained three cohorts: two cohorts of patients with PSC and one cohort of healthy controls. The first PSC cohort used 30 serum samples of patients with PSC from the UCL Institute for Liver and Digestive Health, Royal Free Hospital, UK. The second PSC cohort used 16 predose serum samples taken from patients with PSC enrolled in the ongoing SPRING phase 2a study (NCT04595825) of CM-101 in patients with PSC. The 30 samples of healthy controls were taken from the predose serum samples from a phase 1 study (protocol number CM-101-I-001; CRC number TRC 085/10245). Sample collection followed the ethical principles outlined in the Declaration of Helsinki and received approval from participating centers. All participants provided written informed consent prior to sample collection.

Demographics and clinical parameters such as age, sex, serum biochemistry, blood counts, and evidence of malignancy or disease-related clinical manifestations were also tabulated for these patients. The Enhanced Liver Fibrosis (ELF)™ score was available for 20 out of 30 subjects in the first PSC cohort and for 14 out of 16 patients in the second PSC cohort.

### 5.2. Human Serum Proteomics Assay

The Olink^®^ Explore 3072 platform was used to measure serum levels of 2926 unique proteins by proximity extension immunoassay (PEA; Olink^®^ Proteomics, Uppsala, Sweden), as described previously [12]. Briefly, PEA technology allows for recognition of proteins in the serum by antibody pairs attached to cDNA strands and detected by PCR array, resulting in a normalized protein expression value (NPX), a log-2 scale matrix of relative quantification levels.

Data were read into R (v.4.3.0), and several identifiers were recoded; one patient was excluded from all downstream analysis following abnormal values in many protein values, likely due to sample hemolysis. This exclusion was based on principal component analysis (PCA) using the Olink^®^ Analysis R package (v.3.4.1). All other values were kept, including those with NPX values lower than the limit of detection or with QC warnings. We limited our data set to proteins expected to be found in the serum. To do this, two protein lists were queried: the intracellular proteins list by the human protein atlas [34] was used for exclusion and the explore 1536 panel was used for inclusion, leaving 2072 unique proteins for downstream analysis.

Data analysis was based on three stratifications: (1) disease state (patients with PSC vs. healthy controls; n = 30 and 45), (2) fibrosis state (patients with PSC stratified by ELF™ score, 9.8 threshold [13,14]; n = 15 and 19 for low and high, respectively), and (3) CCL24 serum level (patients with PSC stratified by median NPX CCL24 value; n = 22 and 23 for low and high, respectively). When comparing high and low expression of the three eotaxins (CCL11, CCL24, CCL26), stratification is performed by the mean NPX value of that protein in the PSC patient population (n = 20, 25, 24, 21, 24, and 21 for low CCL24, high CCL24, low CCL11, high CCL11, low CCL26, and high CCL26, respectively).

Ingenuity pathway analysis (IPA^®^, Qiagen, Dusseldorf, Germany) was employed for the analysis using differentially expressed proteins (DEPs, *p* value of 0.05) tested for each stratification (disease state, fibrosis state, and CCL24 level). Data were entered as type “Expr Other”, and default analysis settings were used. For characterization of liver-related toxicity functions, the toxicity functions were filtered to those related to the liver tissue.

For assessment of elevation in neutrophil and monocyte recruitment pathways, the M34121, M12319, M16250, and M4956 gene sets were used.

### 5.3. Cell Culture

The hepatic stellate cell line LX2, catalog number #SCC064, was obtained from Millipore and was seeded at 1 × 10^6^ cells/well in 6-wells plate in growth medium (DMEM, 1% Pen-Strep-glutamine, 2% FBS); after 24 h, medium was replaced with starvation medium (DMEM, 1% Pen-Strep-glutamine, 0.5% FBS) for 24 h. Medium was removed and cells were treated in serum-free media for 48 h with PBS, CCL24 25 ng/mL, CCL24 25 ng/mL + CM-101 2.5 µg/mL, and CCL24 25 ng/mL + CM-101 5 µg/mL. The culture medium was collected and subjected to a RayBiotech L507 protein array to assess the level of secreted biomarkers in each treatment group.

### 5.4. Animals

C57bl/6 male mice, 12 weeks old, and BALB/C male mice, 6 weeks old, were purchased from Envigo (Ness-Ziona, Israel) and were acclimatized for 7 days at the animal house. For mice experiments, we used the human/mouse cross-reactive CM-101 antibody with a mouse Fc-portion.

For intraperitoneal (i.p.). immune cell trafficking studies, mice were randomized into groups of 4–5 in each experiment. Mice were subjected to s.c. injection of either vehicle, 100 μg isotype mouse IgG1, or 100 μg CM-101, followed by 200 μL i.p. injection of either PBS, 5 μg CCL24, or 5 μg CCL11. Four hours after treatment, peritoneal fluid was collected and cells were incubated with FACS-lysing solution, centrifuged, and washed. Animal work was performed following approval of the “National Board of Animal Studies in the Ministry of Health” by the Hadassah Medical Center.

For the α-naphthylisothiocyanate (ANIT) diet model, male, 6-week-old wild-type (WT) C57BL/6 mice were fed 2018 Teklad Global Rodent supplemented either with or without 0.05% ANIT for 4 weeks. ANIT-fed mice were i.p.-injected twice weekly with PBS or 5 mg/kg CM101 (7 animals per group) from week 2 until week 4. A wild-type control group contained 4 mice. Mice were sacrificed after 4 weeks. Animal work was performed following approval of the “Board of Animal Studies in the Ministry of Health” (No-IL-18-6-179), Kaplan Medical Center.

Mdr2^−/−^ mice (male and female, C57BL/6J background) were bred and tested in Hadassah Medical Center. CM-101 treatment was previously described [11]. Animal work was performed following approval of the “National Board of Animal Studies in the Ministry of Health” No-MD-18-15651-2 by the Hadassah Medical Center.

### 5.5. Flow Cytometry of Mouse Peritoneal Immune Cells

Peritoneal fluid samples were stained with fluorochrome-conjugated anti-mouse monoclonal antibodies: CD11b-APC (101212, BioLegend, San Diego, CA, USA), CD3-PE (100308, BioLegend, San Diego, CA, USA), Ly6C-PE (128007, BioLegend, San Diego, CA, USA), and Ly6G-FITC (127605, BioLegend, San Diego, CA, USA). Samples were analyzed using the CytoFLEX flow cytometer (Beckman Coulter, California, CA, USA). The CD11b+ leukocytes were gated by Ly6C/Ly6G into neutrophils (Ly6C^med^/Ly6G^+^) and monocytes (Ly6C^high^/Ly6G^−^). T cells were gated by CD3.

### 5.6. scRNA-seq of Mouse Peritoneal Immune Cells

Raw reads of each sample were processed using the “count” command of the 10× Genomics Cell Ranger software, v2.0.2, aligning the reads to the mouse mm10 (GRCm38) genome. The generated report was used for assessing the quality of the samples. Samples were analyzed by Seurat 3.0.2. Data sets were normalized using “LogNormalize”, a global-scaling normalization method.

### 5.7. Immunohistochemistry

Mouse liver tissues were trimmed, fixed in 4% neutral buffered formalin, embedded in paraffin, and sectioned at 4 μm thickness. Slides were baked for 120 min at 60 °C, dewaxed, and pretreated with epitope-retrieval solution (ER, Leica Biosystems Newcastle Ltd., Newcastle Upon Tyne, UK), followed by incubation for 30 min with primary antibodies. Detection of CD3 (Abcam, Cambridge, UK, #Ab5690, 1:200 concentration) and Iba1 (Abcam, #Ab178847, 1:2000 concentration) was performed using the Leica Bond Polymer Refine HRP kit (Leica Biosystems Newcastle Ltd., Newcastle Upon Tyne, UK), while Ly6G (R&D Systems, Minneapolis, MN, USA, #MAB10371, 1:400 concentration) was detected using an anti-rat IgG secondary antibody (VC005 by R&D Systems, Minneapolis, MN, USA). All slides were counter-stained with hematoxylin. Hematoxylin and eosin (H&E) staining was imaged using an Olympus BX60 microscope equipped with a DP-73 camera at magnifications of x4, x10, x20, and x40. Grading for necrosis (by number of inflammatory cells per X20 High Power Field (HPF); 0 = none, 1 = 10–20 per HPF, 2 = 20–50 per HPF, 3 > 50 per HPF), fibrosis (by percentage of fibrosis of the tissue; 0 = 0%, 1 = <25%, 2 = 25–75%, 3 > 75%), and biliary hyperplasia (by number of extra ducts in portal area; 0 = 0, 1 = 2–10, 2 = 10–20, 3 > 20) were analyzed by Patho-Logica.

### 5.8. Analysis of Mouse Serum Biochemistry

Blood samples were collected into serum separation tubes, left at room temperature for 30 min, and centrifuged at 3500× *g* for 10 min at room temperature. The supernatant was collected and stored at −80 °C. Serum levels of liver enzymes and bile acids were measured using Cobas6000 (Roche Diagnostics International, Rotkreuz, Switzerland) and validated using LIRIS software (Zürich, Switzerland).

## 6. Statistical Analysis

Graphics were generated using the R package ggplot2 (v.3.4.2) accompanied by other packages from the tidyverse (v.2.0.0) package collection.

DEPs were calculated by a Welch 2-sample *t*-test at confidence level 0.95 for every protein without false discovery rate (FDR) correction.

Analyses of two groups by *t*-test were performed using R and Prism 9 (GraphPad) software (v.10.1.2). A *p* value of less than 0.05 was considered significant, with the degree of significance indicated on the graphs. No multiple-testing correction was employed.

## 7. Use of Generative Al in the Writing Process

During the preparation of this work, the authors used ChatGPT in order to improve readability. After using this tool, the authors reviewed and edited the content as needed and take full responsibility for the content of the publication.

## Figures and Tables

**Figure 1 cells-13-00209-f001:**
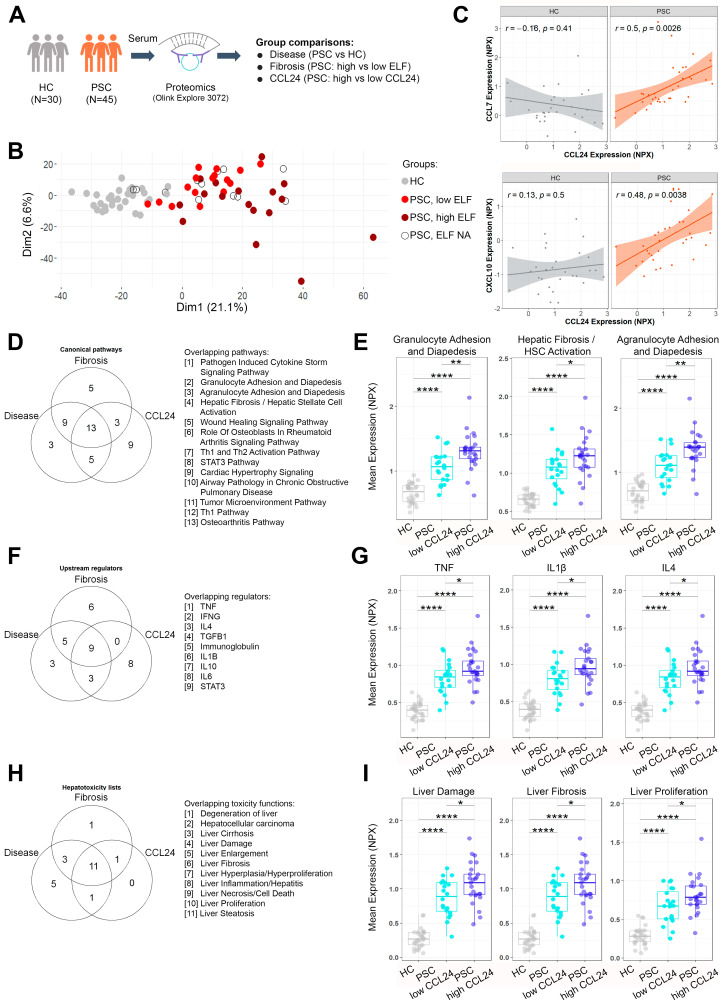
PSC-related mechanisms are upregulated in patients with high CCL24 levels. (**A**) Analysis overview: sera from patients with PSC and HC were analyzed by the Olink Explore 3072 proteomic platform, and differentially expressed proteins (DEPs) were compared by disease, fibrosis, or CCL24. (**B**) Score plots of principal component analysis of proteome profiles in HC and patients with PSC with low or high ELF scores. (**C**) Correlation of CCL24 with representative proteins associated with inflammation/chemotaxis (CCL7 or CXCL10) in HC (n = 30) or in patients with PSC with ALP > 1.5 ULN (n = 34). (**D**–**I**) Ingenuity pathway analysis of canonical pathways, upstream regulators, and liver-related toxicity functions. Venn diagrams show top 30 significant canonical pathways (**D**), top 20 significant upstream regulators (**F**), and significant liver-related toxicities (**H**). The average expression of protein lists of specific canonical pathways (**E**), upstream regulators (**G**), and liver-related toxicities (**I**) are presented for HC and patients with low and high CCL24 serum levels. Boxes represent interquartile ranges with medians (n = 20–30). * *p* < 0.05; ** *p* < 0.01; **** *p* < 0.0001. ELF, Enhanced Liver Fibrosis; HC, healthy controls; NA, not assigned; NPX, normalized protein expression.

**Figure 2 cells-13-00209-f002:**
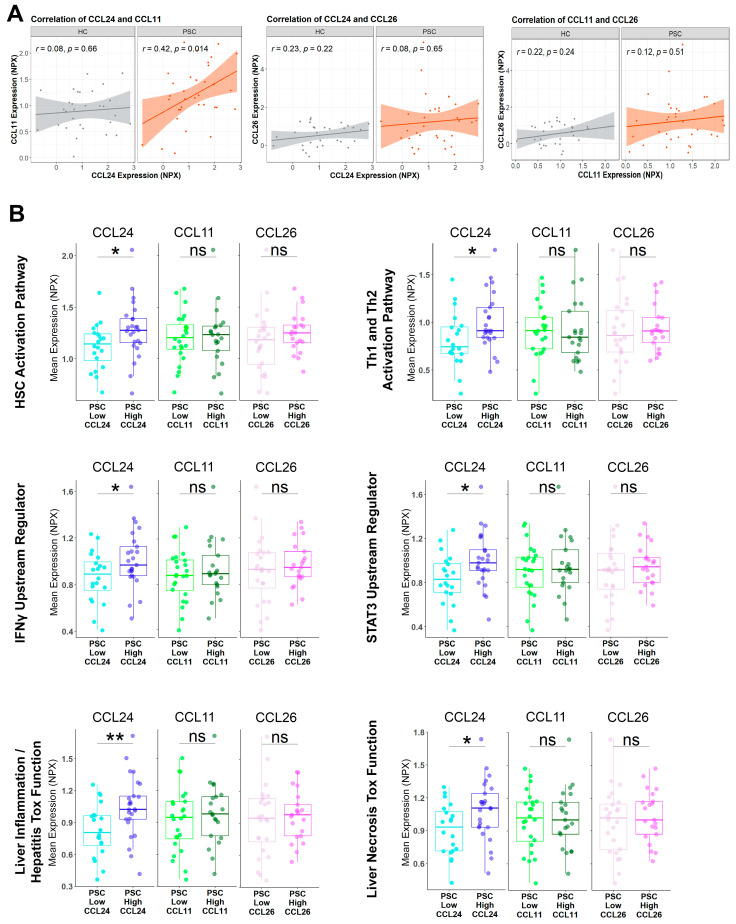
PSC-related mechanisms are specifically associated with CCL24 and not with other eotaxins. (**A**) Correlation of CCL24, CCL11, and CCL26 in HC (n = 30) or patients with PSC with ALP > 1.5 ULN (n = 34). (**B**) The average expression of protein signatures was examined in patients with PSC, stratified by mean expression of each of the three eotaxins, CCL24, CCL11, and CCL26. Boxes represent interquartile ranges with medians (n = 20–25). * *p* < 0.05; ** *p* < 0.01, ns, no significance.

**Figure 3 cells-13-00209-f003:**
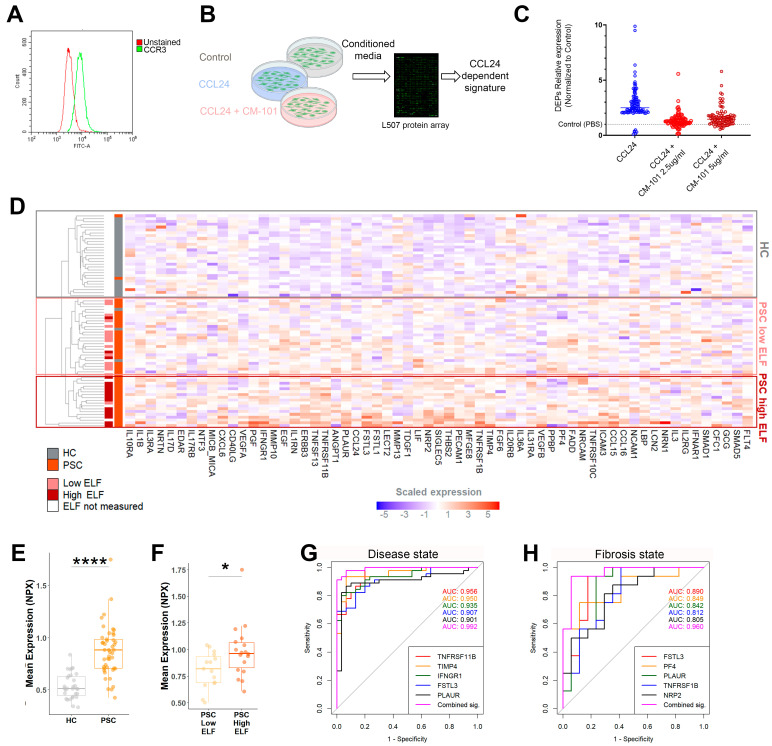
Hepatic cell line CCL24-dependent secreted proteins can differentiate individuals by PSC and its severity. (**A**) FACS analysis of LX-2 cells stained for CCR3. (**B**) Outline of the experimental procedure. LX2 cells were incubated with either PBS, CCL24, or CCL24 with CM-101. (**C**) Relative expression of the CCL24-dependent protein signature. (**D**) Differences in the serum proteomic profile of the CCL24-dependent signature were examined by an unsupervised heatmap of scaled expression values for each individual and identified protein. (**E**,**F**) The average expression of CCL24-dependent protein signatures is presented in HC vs. patients with PSC (**E**) or in patients with PSC, stratified by ELF score of 9.8 (**F**). Boxes represent interquartile ranges with medians (n = 20–30). (**G**,**H**) The ROC curves for the five serum proteins with the highest AUC values for predicting disease presence (**G**) or fibrosis severity (**H**). * *p* < 0.05; **** *p* < 0.0001.

**Figure 4 cells-13-00209-f004:**
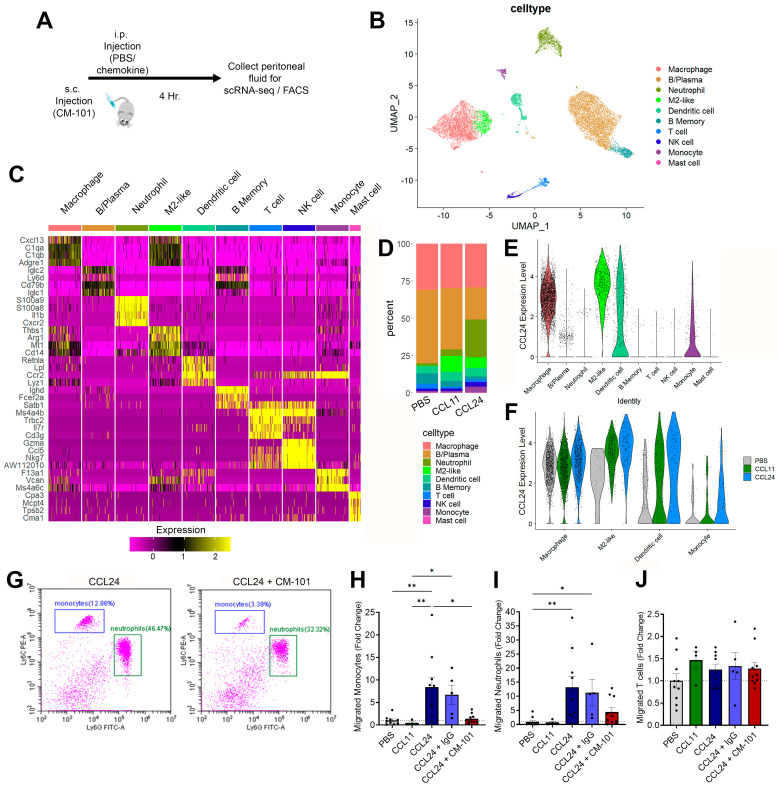
CCL24 i.p. injection recruits neutrophils and monocytes. (**A**) Outline of the experimental procedure. (**B**–**F**) Identification of peritoneal immune populations by scRNA-seq. (**B**) Uniform manifold approximation and projection (UMAP) plot of all cells that passed QC, clustered by the Louvain algorithm. (**C**) Heatmap of the top four gene markers for each cluster. Expression levels are scaled, and the number of cells in each column is downsampled to 500. (**D**) Stacked bar plot showing the relative abundance of each cell type, per treatment. (**E**) Violin plots of CCL24 expression levels by cell type. (**F**) Violin plots of CCL24 expression levels by cell type and treatment. (**G**–**J**) Identification of peritoneal immune populations by flow-cytometry. (**G**) Representative scatter plots of CD11b+ cells. (**H**–**J**) Quantification of relative change in cell counts for monocytes (**H**), neutrophils (**I**), and T cells (**J**). Data are represented as mean ± SEM (n = 4–10). * *p* < 0.05; ** *p* < 0.01.

**Figure 5 cells-13-00209-f005:**
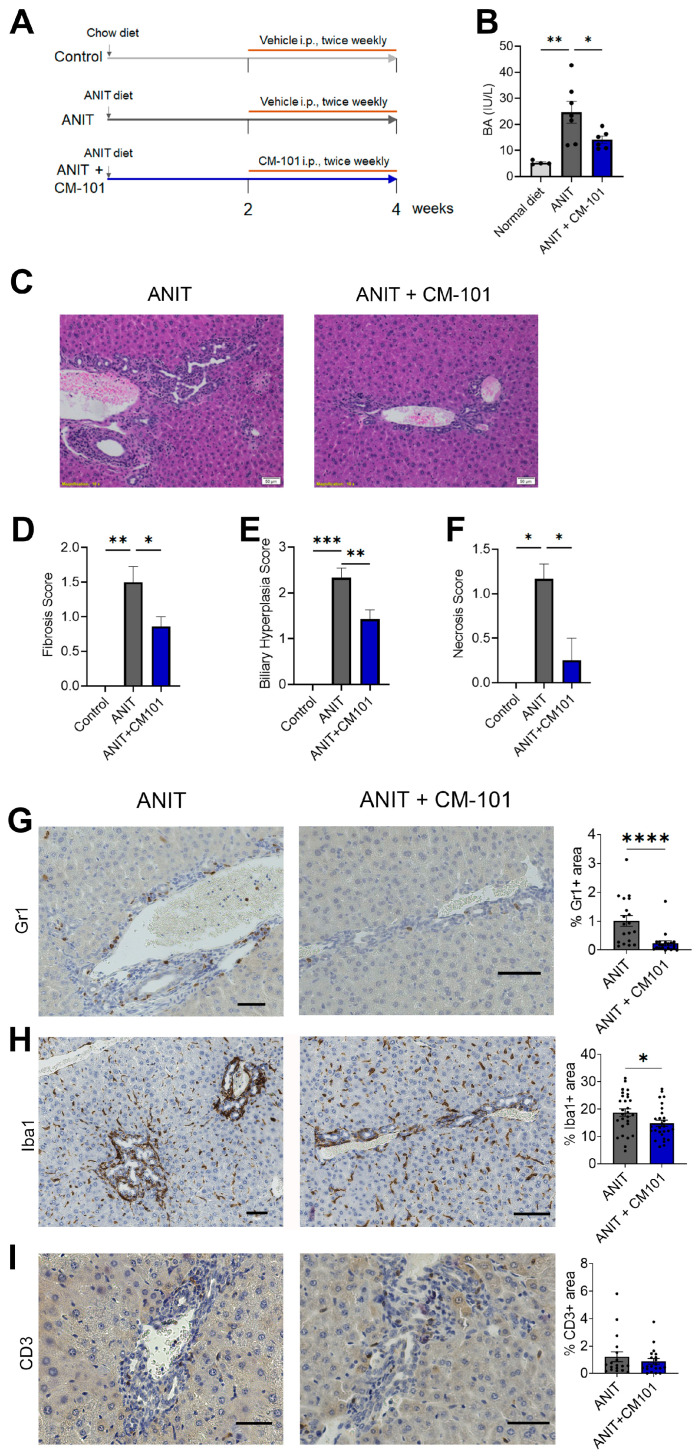
CCL24 blockade attenuates neutrophil and macrophage accumulation and liver fibrosis in ANIT-induced cholestasis model. (**A**) Scheme of the experimental design. (**B**) Levels of serum bile acids (BA) under normal diet, ANIT diet, and ANIT diet with CM-101 (n = 4–7 mice). Data are represented as mean ± SEM. (**C**) Representative H&E staining in liver sections of ANIT diet or ANIT diet with CM-101. (**D**–**F**) Histological scoring of fibrosis (**D**), biliary hyperplasia (**E**), and necrosis (**F**). Data are represented as mean ± SEM (n = 2–7). (**G**–**I**) Representative immunohistochemistry staining and quantification of neutrophils (**G**), macrophages (**H**), and T cells (**I**). Data are represented as mean ± SEM (n = 4–7 mice, 4 fields per mouse). Scalebar represents 50 μm. * *p* < 0.05; ** *p* < 0.01; *** *p* < 0.001; **** *p* < 0.0001.

**Figure 6 cells-13-00209-f006:**
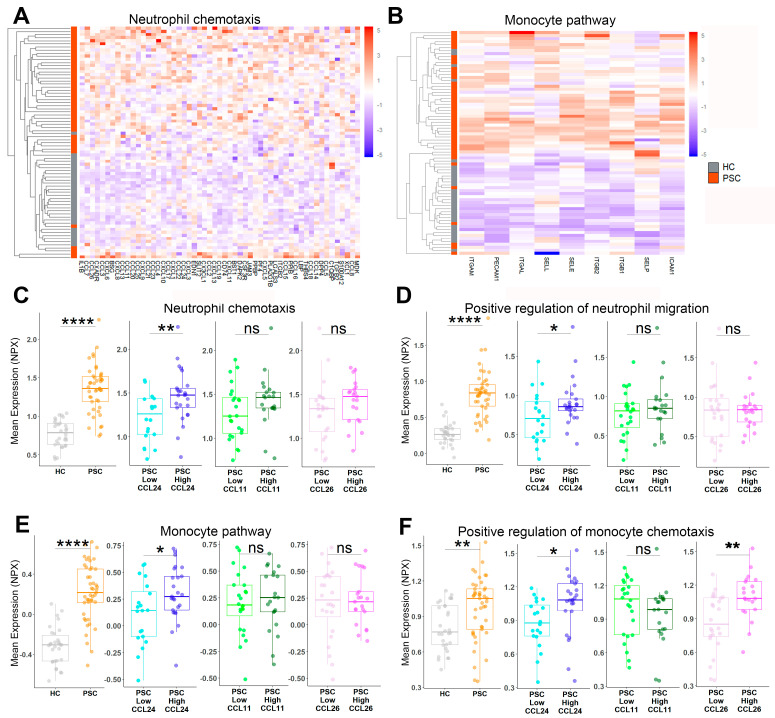
CCL24 is associated with monocyte and neutrophil migration in patients with PSC. (**A**,**B**) Serum proteomic profile of neutrophil chemotaxis and monocyte pathway gene lists were examined by a clustered heatmap of scaled expression values for each individual and protein. (**C**–**F**) The average expression of protein lists of neutrophil chemotaxis (**C**), positive regulation of neutrophil migration (**D**), monocyte pathway (**E**), and positive regulation of monocyte chemotaxis (**F**) are presented for HC and patients with PSC, stratified by mean expression of each of the three eotaxins. Boxes represent interquartile ranges with medians (n = 20–30). * *p* < 0.05; ** *p* < 0.01; **** *p* < 0.0001, ns, no significance.

**Table 1 cells-13-00209-t001:** Clinical characteristics of the study cohorts.

	HC	PSC, Cohort 1	PSC, Cohort 2
N	30	30	15
Age [y], median (range)	23.5 (18–38)	46 (18–76)	37 (23–74)
Duration since diagnosis [y], median (range)	NA	4.3 (0–25.3)	5.1 (0.5–24.5)
Male, n (%)	30 (100)	19 (63)	7 (47)
IBD any, n (%)	0 (0)	19 (63)	11 (73)
ALP [U/L], median (range)	74 (42–106)	235 (52–1064)	355 (273–521)
ALT [U/L], median (range)	16.5 (7–45)	63.5 (20–796)	98 (10–256)
AST [U/L], median (range)	19 (12–34)	48 (15–919)	73 (29–135)
Bilirubin [mg/dL], median (range)	11 (7–18)	13.5 (3–38)	10 (5–41)
Fibroscan [kPa], n, median (range)	NA	30, 10.2 (6.7–17.3)	13, 8.1 (5.0–13.6)
ELF, n, median (range)	NA	20, 10.03 (7.85–11.61)	13, 9.54 (8.49–12.54)

**Table 2 cells-13-00209-t002:** Clinical characteristics of PSC patients divided by CCL24 serum expression.

	PSC, CCL24 Low	PSC, CCL24 High
N	20	25
Age [y], median (range)	40.5 (25–73)	47 (18–76)
Duration since diagnosis [y], median (range)	4.6 (0–25.3)	5.1 (0.1–21.2)
Male, n (%)	15 (75)	11 (44)
IBD any, n (%)	13 (65)	17 (68)
ALP [U/L], median (range)	250 (52–1064)	246 (72–1049)
ALT [U/L], median (range)	91 (10–256)	66 (20–796)
AST [U/L], median (range)	51.5 (20–218)	56 (15–919)
Bilirubin [mg/dL], median (range)	13 (5–41)	12 (3–32)
Fibroscan, n, median (range)	18, 9.6 (5.0–12.6)	25, 10.3 (6.6–13.3)
ELF, n, median (range)	15, 9.95 (7.85–12.54)	21, 9.90 (8.49–11.61)

## Data Availability

The serum proteomics data are not publicly available, as these data are part of an ongoing clinical trial. The data are available from the corresponding author upon reasonable request. The HSC in vitro secretome protein data are available in the published article and its online Appendix A.

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
