# Peer review of "The Role of CCL24 in Primary Sclerosing Cholangitis: Bridging Patient Serum Proteomics to Preclinical Data"

_cells, 2024, doi:10.3390/cells13030209_

Round 1
Reviewer 1 Report
Comments and Suggestions for Authors
Primary sclerosing cholangitis (PSC) is a chronic and progressive liver disease, characterized by fibrosis and peribiliary inflammation. The average age at diagnosis is between 30 and 40 years, with men accounting for about two thirds of cases. As in other autoimmune diseases, genetic, environmental and immune factors are involved in the pathogenesis of PSC, although its etiology is not yet precisely known. Clinically, patients with PSC can be asymptomatic, but typically present cholestasis, recurrent cholangitis, end-stage liver disease and even malignancy. The risk of several malignancies increases in patients with PSC, including cholangiocarcinoma (CCA) and hepatocellular carcinoma. Chemokines play a crucial role in chronic liver disease, contributing to hepatic stellate cells activation (HSCs) and thus promoting tissue fibrosis.
The manuscript by Greenman et al. focuses on the role of chemokine CCL24 in the mechanism of damage and progression of PSC.
The study is divided into three parts. In the first part, a proteomic analysis is carried out on 3 cohorts of statistically relevant patients. In the second part, in vitro cell line studies are performed and in the last part, there are in vivo studies on mouse models.
Having regard to the above, the work is conducted in a coherent, clear and well-structured manner. The abstract graphical is very useful to have an overview of all the work. The paragraph of the methods is well detailed and accurately described.
The results obtained appear consistent with the purpose of the work, and well illustrated.
However some considerations:
· in line 52 the quotations are divided by - instead of the comma, as reported in the rest of the work
· I suggest, in order to make them clearer and readable, to separate Figures 1, 2, 3, and 4 into separate figures
· good level of English
· there are no self-references and the bibliography is recent.
Author Response
We thank the reviewer for his comments on our manuscript, and offer the following response to each point raised:
- In regard to (1), the quotation marks divided by a hyphen instead of a comma in line 52, this is in accordance with MDPI format for citing several sequential references. We slightly changed the citation in question for references 6, 7, 8, and 9, and it will now appear as (6-9).
- In regard to (2), rearranging figures 1, 2, 3, and 4 and in accordance with comments received from other reviewers, we made several changes to figures throughout the manuscript and the supplementary material.
- We thank the reviewer for his comments on the written language and references in points (3) and (4)
Reviewer 2 Report
Comments and Suggestions for Authors
Greenman R et al investigated the role of CCL24 in human primary sclerosing cholangitis (PSC) in a comprehensive manner, with proteomics analysis of patient serum and several preclinical models, the latter of which were ranged from a cell-based assay to an in vivo cholestasis model in mice. Over all, the manuscript that was a translational extension/complementary study from the authors’ previous finding (JCI, 2023), was well conceived, analyzed, and presented, especially for the scientific community studying notoriously intractable PSC.
Major Concern
1. In Figure B: Could hierarchical clustering with all 2081 protein expression levels separate PSC patients into several distinct clusters? Dichotomous separation of PSC patients with a clinically prescribed ELF 9.8 should be meaningful, but an exploratory clustering of PSC patients with regard to serum proteomics may give us a hint for disease pathogenesis/subtypes.
2. In Figure 1C: Authors should comment the reason to choose CCL7 and CXCL10 as variables for correlation with CCL24.
3. Supplementary Table S2-7 and 9 were missing.
4. In Figure 4B, D: Weren’t there M1-like macrophages in peritoneal cavity (4B)? No statistical analysis was depicted (4D).
5. In Figure 6: With regard to the relative expression levels of each protein in the signatures of monocyte and neutrophil related pathways in patient serum, could simple hierarchical clustering differentiate ELF-high from ELF-low PSC patients with high diagnostic value?
Author Response
We thank the reviewer for his comments on our manuscript, and offer the following response to each point raised:
- In regard to (1), the suggestion that hierarchical clustering could help identify subtypes of patients with PSC, while such clusters are created (see below) and show some grade of separation, we think that an analysis relaying on such a large number of proteins to define disease subtypes will not be sufficiently precise.

- In regard to (2), the selection of CCL7 and CXCL10, we agree that it should be further explained, and the text was changed accordingly (lines 79-80). These proteins are representative of a larger group of proteins, each showing a similar pattern of differentiated correlation between HC and PSC patients.
- In regard to (3), the missing supplementary tables, all tables are now provided in a single XLS file, with each worksheet corresponding to a table.
- In regard to (4), the missing M1-like macrophages in the single cell RNA data, when annotating the clustered results of the UMAP, no canonical indicators of M1-like macrophages were observed. The complete list of markers can be found in supplementary table 7. We hypothesize that this is the case since the two treatments (CCL24 and CCL21) are associated with a type 2 immune response, by induction of migration of type 2 immune cells, such as Th2. Regarding the statistical analysis, the single-cell RNAseq was performed on pooled samples from several mice, so no statistical testing was appropriate.
- In regard to (5), the addition of clustering to the heatmaps in figure 6 to see if patients with PSC would be differentiated for their ELF levels, we tested this and unfortunately, we do not see separation of ELF levels, only of PSC/HC. We agree that the heatmaps should include clustering and have replaced the current heatmaps in figure 6.
Reviewer 3 Report
Comments and Suggestions for Authors In this study, the authors clearly showed that CCL24 has important roles in the development and progression of primary sclerosing cholangitis. The experiments are well designed, and the data are robust. This manuscript, overall, has significant merit. Nonetheless, there are several concerns that need to be addressed before considering its acceptance.1. Some of Supplementary Tables are missing.
2. Abstract, line 209: ANIT should be defined.
3. Figure 1DF: It should be discussed whether CCL24/CCR3-related pathway or regulators are predicted.
4. Figure S1C: Please add the pathways shown in Figure 1E (7 pathways in total).
5. Figure S1D: Please add the Th1/Th2 activation pathway.
6. Figure S1EF: Please show the 7 pathways.
7. Figure S1DEF: HC samples should be included.
8. Figures 1F&S2: Because these regulators, except for STAT3, are serum proteins, the serum levels of these regulators determined by the proteomics analysis should be shown.
9. Figure S2: IFNG in high or low CCL24/CCL11/CCL26 should be shown.
10. Figure S2BCD: HC samples should be included.
11. Figure S3: As indicated above, please be consistent with Figure 1HI.
12. Figure 2B: These data are mostly same as shown in Figures S1-S3, and somewhat confusing because of different pathways/regulators. Please be consistent with Figure 1EGI. For more simplification, Figure 2B could merge with Figures S1-S3.
13. Figure S2E: In addition to Figure S2E, Serum CCL24/11/26 levels in HC and PSC should be shown.
14. It seems confusing that CCL24 induces immune cell infiltration/mobilization, I am wondering whether CCL24 directly affects HSCs in the pathology of PSC. Therefore, CCR3 expression in LX2 cells and/or liver section from model mice should be examined.
15. In general, M2 macrophages suppresses immune response. Please discuss their pathological roles in PSC.
16. Although T cell-related pathways are predicted in Figure 1D, it is suggested that those cells are not involved in the mechanism of action of CM-101. This point (including the involvement of T cells in the pathology of PSC) should be discussed.
17. Figure 6AB: Please discuss whether these pathways and their component proteins are overlapped with granulocyte/agranulocyte-related pathways (Figure 1E).
Author Response
We thank the reviewer for his comments on our manuscript, and offer the following response to each point raised:
- In regard to (1), the missing supplementary tables, all tables are now provided in a single XLS file, with each worksheet corresponding to a table.
- In regard to (2), definition of ANIT, we replaced the abbreviation with the full name in line 24. ANIT is later defined in the text as in the abbreviation section of the manuscript.
- In regard to (3), prediction of pathways based on CCL24/CCR3 interference, the following text was added in line 297: “Indeed, the analysis predicted known regulators influencing the CCL24/CCR3 axis, such as IL4, IL10, STAT3 and IFNγ”.
- In regard to (4, 5, 6, 7, 9, 10, 11, 13), we agree that all plots should be consistent throughout the supplementary materials. We have changed supplementary figures S1-S3, and divided them into four supplementary figures (supplementary figures S1-S4). We believe the revised supplementary figures S2-S4 allow for a more uniform and coherent interpretation of the plots, we thank the reviewer for taking the time to comment on this.
- In regard to (8), we have looked at these serum proteins for each group (HC, PSC with lower or higher than average CCL24). All proteins, except IFNG and IL6, show the expected pattern of increased expression as CCL24 levels increase. We have added a figure to reflect this analysis to the supplementary materials, figure S1C.
- In regard to (12), our aim was to focus on the comparison of eotaxins 1, 2 and 3 in patients with PSC in figure 2. With the revisions to the supplementary materials, which now consistently includes all pathways, we believe figure 2 in its current form would make sense.
- In regard to (14), CCR3 expression on LX2 was added to fig. 3, and the text was changed accordingly in line 132.
- In regard to (15), the role of M2 macrophages in the immune response, the following was revised in the discussion, starting from line 314: “Accumulation of peribiliary monocyte-derived macrophages was shown to be a feature of PSC, including M2-like macrophages, which promote fibrosis and secrete CCL24”.
- In regard to (16), CM-101’s mechanism of action and the involvement of T cells, the following statement was added to the discussion, starting in line 333: “The in-vivo investigations conducted in this study highlight the mobilization of myeloid cells, specifically monocytes and neutrophils, without demonstrating a discernible impact on lymphoid cells. Nonetheless, the proteomic analysis revealed heightened activation of both Th1 and Th2 pathways in patients with elevated CCL24 levels, implying that the murine model might not capture the full scope of CCL24 mode of action in the pathogenesis of PSC”.
- In regard to (17), the overlap of select pathways, we found small overlap and therefore did not discuss this point specifically. For example, neutrophil chemotaxis and granulocyte adhesion and diapedesis shared ~34% of the proteins listed, positive regulation of neutrophil migration and granulocyte adhesion and diapedesis shared ~15%, while positive regulation of monocyte chemotaxis and agranulocytes adhesion and diapedesis shared ~19%.
Round 2
Reviewer 2 Report
Comments and Suggestions for Authors
Authors responded appropriately to Review's comments.
Reviewer 3 Report
Comments and Suggestions for Authors
The authors have sufficiently addressed my comments and successfully improved the manuscript.